# Building Vulnerability to Landslides: Broad-Scale Assessment in Xinxing County, China

**DOI:** 10.3390/s24134366

**Published:** 2024-07-05

**Authors:** Fengting Shi, Ling Li, Xueling Wu, Yueyue Wang, Ruiqing Niu

**Affiliations:** 1School of Geophysics and Geomatics, China University of Geosciences, Wuhan 430074, China; sft1459341072@gmail.com (F.S.); snowforesting@163.com (X.W.); yyw@cug.edu.cn (Y.W.); 2School of Future Technology, China University of Geosciences, Wuhan 430074, China; liling@cug.edu.cn

**Keywords:** landslide hazard, building resistance, CNN, quantitative risk modeling

## Abstract

This study develops a model to assess building vulnerability across Xinxing County by integrating quantitative derivation with machine learning techniques. Building vulnerability is characterized as a function of landslide hazard risk and building resistance, wherein landslide hazard risk is derived using CNN (1D) for nine hazard-causing factors (elevation, slope, slope shape, geotechnical body type, geological structure, vegetation cover, watershed, and land-use type) and landslide sites; building resistance is determined through quantitative derivation. After evaluating the building susceptibility of all the structures, the susceptibility of each village is then calculated through subvillage statistics, which are aimed at identifying the specific needs of each area. Simultaneously, different landslide hazard classes are categorized, and an analysis of the correlation between building resistance and susceptibility reveals that building susceptibility exhibits a positive correlation with landslide hazard and a negative correlation with building resistance. Following a comprehensive assessment of building susceptibility in Xinxing County, a sample encompassing different landslide intensity areas and susceptibility classes of buildings was chosen for on-site validation, thus yielding an accuracy rate of the results as high as 94.5%.

## 1. Introduction

Vulnerability is characterized as the degree of loss sustained by a given element or group of elements within an area affected by a disaster, and it is quantified on a scale from 0 (no loss) to 1 (total loss) [1]. Existing studies categorize the methods for assessing building vulnerability into two primary categories: One approach relies on the investigation and analysis of data from historical disasters, thus providing empirical values or empirical vulnerability curves for different types of buildings by considering their attributes and characteristics; these values typically range between 0 and 1 [2]. The other method considers building vulnerability as the potential loss to a structure, which is determined by calculating the likelihood of exposure to a disaster and the building’s value.

The early research on landslide susceptibility analysis was pioneered by scholars who assessed the susceptibility of landslide hazards using topographic and geological maps. The development of Remote Sensing (RS) and Geographic Information System (GIS) technologies has greatly contributed to the methodology for assessing landslide susceptibility and risk. For instance, the spatial overlay analysis of landslide distribution maps, slope maps, and geological maps has been utilized to enhance the precision of landslide hazard assessments. The introduction of multifactorial integrated prediction methods and the application of GIS technology in data processing and management have further advanced landslide hazard risk assessment methodologies. In recent years, various advanced modeling techniques have been utilized to enhance the prediction and assessment of natural disasters. For example, Xie et al. [3] proposed a Monte Carlo method to estimate the probability of cyclone impacts using historical data, thus providing a probabilistic framework for urban planning and disaster preparedness. Similarly, an integrated hydrological–geotechnical model (iHydroSlide3D v1.0) has been developed to simulate flood–landslide cascading events, thus demonstrating the importance of coupling hydrological and geotechnical processes for accurate disaster prediction [4].

With advancements in computer technology and data processing capabilities, research methods have shifted from qualitative to quantitative analyses and integrated evaluation approaches. Buckle developed a multidimensional social life vulnerability assessment framework [5]. Papathoma-Köhle proposed a quantitative formula for assessing building vulnerability, which influenced by five factors: the building materials, the surrounding environment, the orientation towards the slope, the presence of warning signs, and the number of building floors [6]. Uzielli redefined vulnerability within quantitative risk estimation as a function of landslide strength and the susceptibility of vulnerable elements (V=I∗S), thus providing a detailed explanation [7]. Li’s quantitative model for structural and human susceptibility considers both landslide strength and the resistance of exposed units, thus defining susceptibility (*V*) as a function of hazard intensity (*I*) and the resistance to the threat (*R*) [8]. Furthermore, Silva proposed a semiquantitative approach, thus defining susceptibility as a function of building resistance (BR) and landslide magnitude (LM), with the building resistance weighted by expert knowledge and a landslide magnitude set at 0.6 [9]. In a Portuguese municipality, Guillard-Gonçalves conducted a semiquantitative physical susceptibility assessment of buildings by calculating a weighted average based on the number of each structural building type within the BGRI [10]. Singh expanded on this by defining susceptibility as a function of landslide intensity and building resistance (PV=f(I,R)) [11], thus noting a different approach from Li, where landslide intensity (*I*) was based on landslide volume (*v*) and velocity (*s*). Glade and Crozier highlighted the complexity of quantitatively expressing susceptibility, thus emphasizing the challenge in understanding the relationship between landslide hazard characteristics and damage traits [10]. In 2012, Papathoma-Köhle introduced the concept of a susceptibility curve defined as a function (f(DL,I)) of the process intensity and degree of loss [12]. Kang and Kim developed physical susceptibility curves for different building structures to assess mudslide risk based on empirical data on mudslide intensity and building damage levels [13]. Godfrey performed a physical susceptibility assessment for hydrometeorological hazards using an expert methodology in areas with limited hazard data, thus utilizing susceptibility indices and curves to generate area-specific susceptibility curves [14]. These curves were then applied to assess the mudslide risk across various building structures.

Since machine learning not only reduces human error but also possesses the ability to process big data, as well as enhance accuracy, efficiency, and adaptability, these advantages together contribute to the widespread use of machine learning in the field of building susceptibility assessments. Dai effectively simulated slope instability using GIS techniques and logistic multiple regression analysis [15,16]. Duman used an extensive landslide database and logistic regression analysis to develop a model that characterizes landslide susceptibility in map form, thus achieving an accuracy rate of 83.8% in classifying landslide susceptibility zones [17]. Using GIS and the information value method, maps were produced and classified through geostatistical analysis based on the calculated hazard information values of different evaluating genes [18]. Mück applied remote sensing techniques to assess building vulnerability to earthquakes and tsunamis, thus classifying and mapping this vulnerability [19].Hadji utilized raster-based GIS and statistical processing to delineate landslide hazards in the Souk Ahras province. Tabular data, maps, and satellite images were compiled, processed, and integrated into a spatial database on a GIS platform, with landslide hazard zones being assessed and mapped through the application of probabilistic and logistic regression methods involving maps of landslide incidence and permanence factors [20]. Geiß conducted a quantitative evaluation of the applicability of multisensor remote sensing techniques for assessing the seismic susceptibility of buildings [21]. Thennavan employed ArcGIS to examine building vulnerability and its spatial distribution in the mountainous areas of the Nilgiris district, Western Ghats, India [22]. Techniques such as the Least Squares Support Vector Machine (LSSVM) and Multiclass Alternating Decision Tree (MADT) were utilized to compare methods for the spatial prediction of landslides [23]. Chen employed GIS-based models, including DS, LR, and ANN, to map landslide susceptibility in the Shangzhou district of Shangluo city, Shaanxi Province, China. The results indicate that the ANN model achieved higher training and prediction accuracy compared to the other models [24]. Hong utilized Logistic Regression and Random Forest models to construct landslide susceptibility maps in the Wuyuan area, China [25]. Su et al. employed Support Vector Machines (SVMs), Logistic Regression (LR), and Artificial Neural Networks (ANNs) to develop three distinct landslide models and compare various nonlinear landslide susceptibility mapping techniques [26]. Liu quantified the susceptibility of railway systems to rainfall-induced hazards using Random Forests [27]. Arabameri employed four methodologies—EBF, FR, TOPSIS, and VIKOR—to create flood hazard susceptibility maps (FHSMs), with the EBF model demonstrating a superior prediction rate in the validation phase [28]. Chen evaluated the effectiveness of landslide susceptibility mapping by comparing the traditional bivariate weight of evidence (WoE) with a hybrid approach of multivariate logistic regression (WoE-LR) and machine learning-based Random Forest (WoE-RF), thereby finding that the hybrid model surpassed the others in performance [29]. Zhang employed a Bayesian network model to quantitatively assess earthquake-induced landslide susceptibility, thus considering key factors such as sediment thickness, building damage, evacuation, and resistance [30].

In previous studies on building vulnerability, methods integrating quantitative formula derivation and machine learning have been scarce; most have relied solely on either quantitative formula derivation or pure machine learning for vulnerability assessment. However, these approaches each have their limitations: quantitative formula derivation may not fully utilize modern data analysis techniques, while pure machine learning methods can be overly dependent on large-scale, high-quality data and may lack the ability to be interpreted by humans.

This study addresses these challenges by integrating quantitative formula derivation with machine learning techniques to enhance the accuracy and efficiency of building vulnerability assessments. This integration improves interpretability and reduces the dependence on extensive high-quality data. Specifically, this research aims to develop an effective, accurate, and interpretable model for assessing building vulnerability on a large scale. This study utilizes advanced machine learning algorithms, such as Convolutional Neural Networks (CNN), to quantitatively evaluate landslide hazard risks. It calculates building resistance through quantitative derivation based on structural characteristics, integrates these assessments to evaluate the vulnerability of all buildings in Xinxing County, and validates the model through on-site verification to ensure accuracy and reliability. This approach offers a comprehensive and adaptable framework for assessing vulnerability across diverse scenarios and needs. Furthermore, the paper details the methodology for the landslide hazard assessment, data acquisition, and processing methods, as well as the calculation of building resistance, and it presents the results of the vulnerability assessments and their validation. This comprehensive approach allows for a nuanced understanding of building vulnerability in the context of landslide hazards and offers valuable insights for disaster risk management.

## 2. Proposed Methodology for Quantitative Assessment of Building Vulnerability

Building on previous research in the quantitative assessment of building vulnerability [8,11], this study proposes a formula for the large-scale vulnerability assessment (PV) of buildings affected by landslides. Building vulnerability, denoted as PV, is defined in this study as a function of the landslide hazard (*I*) and building resistance (*R*), as illustrated in Equation (Equation 1).
(1)$$PV=f\left(I,R\right)=\left\{\begin{array}{cc} 2\frac{I^{2}}{R^{2}}\; & \frac{I}{R}\leq 0.5\\ 1-\frac{{2(R-I)}^{2}}{R^{2}}\; & 0.5<\frac{I}{R}\leq 1.0\\ 1.0\; & \frac{I}{R}>1.0 \end{array}\right.$$
where we have the following:

*I* represents the landslide hazard, thus ranging from 0 to 1;

*R* represents the building resistance, thus ranging from 0 to 1.

The distinction from previous studies lies in that *I*, in the building susceptibility equations proposed by Li and Singh, represents the landslide intensity. For Li, *I* is influenced by the dynamic intensity factor (Idyn) and the geometric intensity factor (Igem), while for Singh, *I* is determined by the landslide volume (*v*) and the expected velocity of the landslide (*s*). In this study, *I* is innovatively redefined as the landslide hazard, and a Convolutional Neural Network is employed to quantify this hazard as a specific value within the interval from 0 to 1.

The data were collected from various sources, including satellite imagery, geological surveys, and historical landslide records. The collected data were then cleaned to remove any inconsistencies and standardized to ensure uniformity. To address the issue of class imbalance in the dataset, we applied the Synthetic Minority Oversampling Technique (SMOTE).

We employed a Convolutional Neural Network (CNN) to quantify the landslide hazard (*I*). The CNN model was trained on preprocessed data to learn the complex relationships between the input features (such as topography, geological structure, vegetation cover, etc.) and landslide occurrences. The model architecture included multiple convolutional layers with 3 × 3 filters, ReLU activation functions, and max pooling layers to extract and process the spatial features. The output of the CNN model is a probability value between 0 and 1, thus representing the landslide hazard for each building.

Building resistance (*R*) was quantified using six key factors: structural composition, number of floors, construction materials, building age, geographic location, and accessibility. Each factor was assigned a resistance coefficient based on empirical data and expert judgment. The overall building resistance was calculated using the geometric mean of these coefficients to ensure a balanced contribution from each factor.

The vulnerability of each building (PV) was assessed by integrating the landslide hazard (*I*) and building resistance (*R*) using the proposed formula. This formula quantified the vulnerability of over 230,000 buildings in Xinxing County, thus expressing each building’s vulnerability as a specific value between 0 and 1. This detailed quantification allowed for precise vulnerability mapping and targeted risk mitigation strategies (Figure 1).

### 2.1. Methods for Landslide Hazard Assessment

Landslide hazard assessment involves evaluating the potential for landslide occurrences by analyzing various contributing factors such as topography, geomorphology, geological structure, vegetation cover, land-use type, and hydrometeorological conditions. The advent of machine learning (ML) techniques has significantly enhanced the accuracy and comprehensiveness of these assessments by modeling the complex relationships between causative factors and landslide events [31].

Several machine learning models have been effectively applied to landslide susceptibility mapping (LSM), with each demonstrating varying degrees of success. Random Forest (RF), Support Vector Machines (SVMs), and Artificial Neural Networks (ANNs) are commonly used ML models for LSM. These models utilize extensive datasets, including satellite imagery, remote sensing data, historical landslide records, and geographical information systems (GISs), to predict landslide-prone areas with high accuracy [32]. For instance, in a study along the Karakorum Highway in Northern Pakistan, researchers applied RF, Extreme Gradient Boosting (XGBoost), K-Nearest Neighbors (k-NN), and Naive Bayes (NB) models to generate a landslide inventory map. The study categorized landslide areas for validation and training, thus finding that the RF model achieved the highest accuracy and demonstrating its effectiveness in integrating multiple causative factors and producing reliable susceptibility maps [33].

Deep learning (DL) models, such as Convolutional Neural Networks (CNNs), have also been employed to enhance landslide susceptibility mapping. These models are particularly effective in handling large datasets and capturing complex patterns within the data. The combination of DL models with traditional ML techniques has led to the development of hybrid and ensemble models, which often outperform single ML models in terms of their accuracy and reliability [31]. By utilizing advanced ML and DL techniques, researchers can create high-resolution landslide susceptibility maps that provide essential information for disaster management and mitigation efforts. These maps aid policymakers, engineers, and the public in identifying high-risk areas and implementing measures to reduce the impact of landslides on communities and infrastructure.

Through these examples, it is evident that machine learning models have significantly improved the precision and reliability of landslide hazard assessments, thus making them indispensable tools in modern geohazard management.

### 2.2. Convolutional Neural Networks

Convolutional Neural Networks (CNNs) are a type of artificial neural network designed for processing structured grid data, such as images. They have proven highly effective in analyzing visual data due to their ability to capture the spatial hierarchies of features through layers of local connections and weight sharing. CNNs consist of multiple layers, including convolutional layers, pooling layers, and fully connected layers. The convolutional layers apply learnable filters to the input data, which helps in detecting various features like edges and textures. These filters slide over the input image, thus performing an elementwise multiplication and sum with the input data, followed by the application of an activation function such as ReLU (Rectified Linear Unit). Pooling layers, typically max pooling, reduce the spatial dimensions of the feature maps while retaining the most significant information. This reduction helps in decreasing computational complexity and controlling overfitting by progressively reducing the spatial size of the representation. After several convolutional and pooling layers, the network includes fully connected layers. These layers flatten the feature maps and connect every neuron in one layer to every neuron in the next layer, thus enabling the network to make the final classification decisions based on the high-level features extracted by the preceding layers [34].

CNNs are particularly effective for tasks such as landslide susceptibility mapping and building vulnerability assessments due to their ability to handle large datasets and complex patterns. By integrating CNNs into the assessment framework, this study aims to compute landslide hazard values and provide a broad-scale quantitative assessment of the disaster intensity for each building across the entire county, thereby improving the overall evaluation of building vulnerability. For instance, in the practice of LSM in Qingchuan County, Sichuan Province, China, Yi evaluated and compared ANN, 1D-CNN, and RNN, thus finding satisfactory performance in predicting susceptible areas [35]. Similarly, CNNs were used to perform GIS-based landslide susceptibility assessment in the Gorzineh-khil region of northeastern Iran, thus showing superior accuracy, precision, and recall compared to other methods [36].

In the assessment of the landslide hazard risk in Xinxing County, CNNs were utilized to analyze complex datasets, including topography, vegetation cover, and historical landslide occurrences. These networks effectively identify patterns indicating geological instability and provide accurate predictions of potentially hazardous areas.

### 2.3. Building Resistance

Amatruda and his coauthors considered the structural–morphological type, state of maintenance, and strategic relevance as factors influencing building susceptibility in their analysis [37]. Vamvatsikos focused on the characteristics affecting seismic performance for the general building population, including building materials, seismic design, level of detailing, building height, and infill panel configuration [38]. Agliata selected five metrics—structure types, as well as the maintenance status, building orientation, number of floors, and number of openings for direct vulnerability assessment, emphasizing operability, applicability, and low variability [39]. Additionally, the Global Seismic Modelling Initiative identifies key factors such as building height, age, design, and construction quality as primary metrics for assessing building stability.

The six pivotal factors identified for calculating building resistance in Xinxing County, based on the available data and house characteristics, include structural composition (*a*), number of floors (*b*), construction materials (*c*), building age (*d*), geographic location (*e*), and accessibility (*f*). Drawing on a quantitative formula initially proposed by Li and widely utilized in the field of building vulnerability assessment by Papathoma-Köhle and Li et al., the model was specifically extended to include the geographic location and accessibility to better measure the resistance value of buildings in Xinxing County, as delineated in Equation (Equation 2) [8,11]. When they proposed their formulas, Li and Singh et al. conducted empirical data and case studies in their respective articles, thus demonstrating the effectiveness and accuracy of the proposed formulas [8,11].

This formula, which has been extensively applied in building vulnerability assessments, employs the geometric mean to calculate the RRR, thus balancing the influence of each factor. The use of the geometric mean ensures that no single factor disproportionately affects the overall resistance value, thus mitigating the impact of extreme values and providing a more stable and realistic assessment of building resistance. This approach aligns with the methodologies in the literature that emphasize a multifactorial assessment of building resistance, thus making it a robust tool for evaluating the inherent resistance of buildings to various hazards. Furthermore, preliminary field surveys in this study revealed a strong correlation between the calculated resistance values and observed building conditions across various disaster scenarios, thereby affirming the reliability of this formula.
(2)$$R=\sqrt[{6}]{a*b*c*d*e*f}$$

## 3. Case Studies

### 3.1. Study Area

Xinxing County, situated within the jurisdiction of Yunfu City in Guangdong Province, spans a longitude of $111^{\circ }58^{\prime }$ to $112^{\circ }31^{\prime }$ East. The county features a diverse and complex topography that rises from the northeast’s lower elevations to the southwest’s higher altitudes, thus culminating at Tianlu Mountain, which is the highest point at 1250.7 m above sea level. This terrain primarily consists of a mixture of small basins, river valleys, hills, and plateaus, and it is predominantly characterized by hills and mountains. Numerous rivers originate from the high mountains in the south and the hills on the east and west, thereby converging into three major water systems, with most streams feeding into the Xijiang River system of the Pearl River Basin. The hydrological conditions in Xinxing are complex, which are characterized by significant seasonal variations in rainfall. This variability is particularly pronounced during the typhoon season from August to September, which often leads to hydrological disasters such as flooding and waterlogging. Moreover, the area is susceptible to soil erosion, landslides, and other mass wasting events, especially in the upstream areas during the rainy season. The region’s geology is further complicated by folding and fracturing activities, which amplify the susceptibility to landslide hazards. This geological complexity, combined with uneven rainfall distribution, heightens the risk of avalanches, landslides, and mudslides. A depiction of Xinxing County’s geographical location is presented in Figure 2.

### 3.2. Data Acquisition

#### 3.2.1. Landslide Sites and Contributing Factors

Remote sensing was primarily employed to acquire the latest remote sensing data and to conduct remote sensing surveys, which were complemented by ground surveys to investigate the geological conditions conducive to disasters. Additionally, on-site inspections and assessments of changes at landslide hazard locations were performed. Utilizing the results from a detailed 1:50,000 landslide disaster survey and other relevant data, a remote sensing survey was executed to examine the geological conditions predisposing to disasters across an area of 1522.26 km². The landslide slope map derived from this survey is displayed in Figure 3.

In Xinxing County, the primary disaster-causing factors include the topography and geomorphology (elevation, slope, and slope shape), rock and soil body types, geological structure, vegetation cover, watersheds, and land-use type, thus encompassing nine aspects in total. The spatial distribution of these nine influences is characterised by the nine sub-figures a–i in Figure 4. Leveraging the outcomes of previous surveys, the disaster prevention geological conditions were further enhanced and deciphered through remote sensing technology. Specifically, topography, geological structure, vegetation cover, land-use type, and gully characteristics were extracted using remote sensing interpretation methods. Additionally, other information related to disaster-prone geological conditions, such as engineering geological rock groups, water areas, and meteorological data, were derived from existing data analysis. These factors influencing the disaster-prone geological conditions are illustrated in Figure 4. To ensure the accuracy of the landslide disaster survey and to address the specific requirements of the landslide disaster risk census efforts in Xinxing County, a 1:50,000 landslide disaster distribution map was utilized. A raster unit size of 30 m × 30 m was selected as the evaluation unit to express the results of the landslide disaster risk assessment.

#### 3.2.2. Structural Parameters

Data pertaining to buildings in Xinxing County were sourced from the “One House, One Land” rural real estate cadastral survey database of Guangdong Province. This study identified six structural parameters that influence building resistance, specifically the building structure (*a*), number of floors (*b*), materials used (*c*), age of the building (*d*), geographical location (*e*), and building accessibility (*f*). Resistance coefficients for these structural parameters were subjectively assigned based on a qualitative assessment [8], as detailed in Table 1. Additionally, these resistance coefficients were quantitatively computed using Equation (Equation 2) to determine the overall building resistance.

## 4. Research Results and Validation

### 4.1. Results

#### 4.1.1. Landslide Hazard Distribution Map of Xinxing County

In this study, to accurately assess the landslide intensity at each building site, we employed a 30 m × 30 m grid cell approach for landslide susceptibility evaluation, thus dividing the entire county into over 2.42 million grid cells. The assessment utilized nine factors: rainfall, geological structure, NDVI (Normalized Difference Vegetation Index), land-use type, slope shape, relief, distance to roads, slope gradient, and engineering rock group. To balance the dataset, we applied the SMOTE (Synthetic Minority Oversampling Technique) method, thus ensuring that the data input into the CNN model was well-distributed. The CNN model was then used for the landslide hazard evaluation. We carefully selected the CNN parameters to optimize the modeling process. This included adjusting the number of convolutional layers, filter sizes, and pooling operations to enhance the feature extraction and model performance. The parameters were fine-tuned based on the dataset’s characteristics and the computational resources available. We utilized grid search techniques to further optimize the model parameters. Grid search systematically traverses predefined parameter combinations, thus evaluating the performance of each combination to find the optimal settings. For instance, we might adjust the number of convolutional layers, the number and size of filters per layer, the type of pooling layers (such as max pooling or average pooling), and the learning rates. Parameter optimization is an iterative process. After each round of optimization, we adjusted the settings based on the model’s performance on the validation set. This process was repeated until we identified the parameter combination that yielded the best performance. After optimizing the parameters, the best parameter combination was selected for model evaluation, thus resulting in an AUC value of 0.85, which indicates a good level of performance. This high AUC value demonstrates the model’s ability to accurately distinguish between high- and low-risk areas. For visualization, the landslide hazard risk at each sample site was quantified and calibrated between 0 and 1. This risk was then visualized through a color gradient—from green (indicating low risk) to red (indicating high risk). This method enhances the readability of the information and the ability to spatially identify potential threats. A map depicting the distribution of landslide hazards in Xinxing County was produced, thus showcasing these results (refer to Figure 5). By employing this comprehensive and optimized approach, the study provides a detailed and accurate landslide hazard distribution map, which is crucial for quantitatively assessing building vulnerability. This quantitative assessment of landslide risk based on grid cells lays the foundation for the vulnerability assessment of buildings in Xinxing County.

The natural breakpoint method is effective for classifying landslide hazard values by addressing the heterogeneity of environmental factors and sample imbalance, thus enhancing model accuracy and reliability [40]. This method has been widely used in landslide susceptibility studies in Guangdong Province. For instance, in Luhe County, it was employed to categorize landslide susceptibility into five levels, thus determining the spatial probability [41]. In the Beijiang River Basin, it successfully identified clustered shallow landslides following extreme rainfall events [42].

Based on these precedents, this study categorized the landslide hazard values in Xinxing County into four zones—low, medium, high, and extremely high—thus providing a robust framework for hazard assessment.The landslide hazard zoning classifications for Xinxing County are shown in Table 2:Extremely High-Hazard Zone

This zone is predominantly located in the northern parts of Liuzu Town and Taiping Town and the southern part of Xincheng Town. It covers an area of approximately 199.36 km^2^, thus representing 13.21% of the county’s total area. A total of 65 landslides and geological hazards have been recorded in this zone.

High-Hazard Zonev

Situated primarily in the middle regions of Tiantang Town, Dongcheng Town, and Nimcun Town, as well as the southern areas of Taiping Town and Liuzu Town, this zone spans about 396.16 km^2^. It constitutes 26.25% of the county’s total area, with 95 landslide and geological hazards identified.

Medium-Hazard Zone

This zone is located mainly in the north of Dongcheng Town, Lezhu Town, and Hetou Town, thus encompassing approximately 444.44 km^2^. This area accounts for about 29.44% of the county’s total area, with 82 landslide geological hazards noted.

Low-Hazard Zone

The low-hazard zone chiefly covers Ridong Town, the southern part of Hetou Town, and the northern region of Tiantang Town. It has a total area of about 614.95 km^2^, thus making up approximately 25.02% of the county’s total area. Here, 24 landslide geological hazards have been observed.

#### 4.1.2. Building Resistance and Building Vulnerability in Xinxing County

In the results section, this study provides a detailed analysis of the building susceptibility, resistance, and structural properties of the buildings in Hwang Kyung Tin village, which were labeled from *a* to *f*. As shown in Table 3. It is important to note that the factors influencing building resistance and their quantitative relationships have been clearly delineated in Equations (1) and (2). Therefore, no additional explanation of these equations is necessary in this section.

Given the extensive number of buildings in Xinxing County, this study employed village-level statistics to illustrate the vulnerability of the structures effectively. The corresponding results of the village-level building vulnerability assessment are depicted in Figure 6. The building vulnerability index for each village was calculated by aggregating the vulnerability indices of the individual buildings.

In the comprehensive analysis of building vulnerability within Xinxing district, this study has employed the quantile method to categorize building vulnerability into four distinct classes: low, medium, high, and very high. Additionally, the study provides detailed information on the number of villages classified within the two highest vulnerability categories—high and very high—including their respective percentage shares of the total. Figure 7 and Figure 8 further detail the top-20 villages in these categories, thus offering crucial insights into areas of heightened vulnerability within the district.

Conducting the vulnerability assessment at the village level enables a more precise understanding of the specific needs and vulnerabilities of each area. Notably, the villages of She Wai and Hei Tsuen, as illustrated in Figure 7, were found to contain a significant number of buildings categorized under the “high” and “very high” vulnerability classes. However, the overall risk profile for Hing Shing County remains manageable with the implementation of appropriate prevention and mitigation strategies. Specifically, the Village of Sherwood yielded the highest concentration of highly vulnerable buildings, totaling 2748, as depicted in Figure 7. This finding underscores the imperative for enhanced restoration and reinforcement measures. Similarly, the Village of Black Village yielded 2256 buildings rated as highly vulnerable. Although these figures indicate potential risks, a proactive disaster risk management approach is advocated.The spatial distribution of potential risk across the county has been further elucidated through an analysis of the percentage of building vulnerability within each village. As shown in Figure 8, Huangjingtian Village exhibited the highest percentage of high-risk buildings at 94.29%, which was closely followed by Dalang Village at 93.81%. These proportions highlight elevated susceptibility and serve as crucial indicators for strategic planning and resource allocation by the authorities.

In this study, the building vulnerability (PV) and landslide hazard risk were classified using the natural breakpoint method. Additionally, the correlation between the building structural characteristics (*R*) and the building vulnerability (PV) was explored to identify buildings across different hazard intensity levels. The results of these correlation analyses are thoroughly presented in Figure 9. The implementation of this methodology not only equips Xinxing County with an effective tool for assessing and managing landslide hazard risks but also offers valuable insights for other regions facing similar challenges.

This study systematically quantified the number of buildings classified into four designated vulnerability categories—low, medium, high, and very high—under varying landslide hazard conditions. The detailed distribution of these classifications is depicted in Figure 10.

(i)Low Landslide Intensity: The hexagonal box plot reveals a dense concentration of buildings with high resistance factors (greater than 0.6), thus correlating with low susceptibility. Supporting this, the bar chart in Figure 10 shows that 43,047 buildings, or 18% of the total in the county, are situated in areas with low landslide hazard risk and demonstrate low susceptibility (PV≤0.5).(ii)Medium Landslide Intensity: A shift to a higher vulnerability profile is evident in the hexagonal box plot. Correspondingly, the bar chart indicates that 17,478 buildings, approximately 7% of the total, are located in areas of medium landslide hazard risk and are categorized as moderately vulnerable. Additionally, 8399 buildings, or about 3.5%, fall into the high vulnerability category (0.5<PV≤0.9).(iii)High Landslide Intensity: The hexagonal box plot illustrates a significant increase in vulnerability (PV>0.7), even among buildings with notable resistance factors. This increase is substantiated by the bar chart in Figure 10, which highlights that 40,179 buildings, nearly 17% of the county’s total, are highly susceptible in areas of high landslide hazard risk.(iv)Very High Landslide Intensity: The hexagonal box plot displays uniformly high susceptibility across all resistance factors. This observation aligns with the bar chart data, thus indicating that 56,609 buildings, or 24% of the total number of buildings in the county, face very high susceptibility (PV close to 1.0) in areas designated as Very High Landslide Hazard Risk.

The data from the assessment of 238,892 buildings, as illustrated in Figure 9 and Figure 10, highlight the urgent need for a comprehensive, multilayered approach to landslide risk management in Xinxing County. While enhancing the resistance of individual buildings can mitigate vulnerability, especially in areas with low to moderate landslide hazard risk, the profound impacts observed in regions with high to very high landslide hazard risk necessitate broader urban planning reforms. These reforms should include emergency preparedness initiatives and efforts to build community resilience, which are essential to effectively safeguard against landslide hazards.

### 4.2. Validation

During the on-site validation process, each building was meticulously examined regarding its external structural condition, construction materials, state of maintenance, and surrounding topography. Special attention was directed towards buildings predicted to have high susceptibility (high PV), with a focus on identifying signs of landslide impact, such as shifting foundations, wall cracks, and other structural damages.

Conversely, buildings predicted to have low susceptibility (low PV) were scrutinized to confirm their good condition, thus aligning with their low-risk ratings. Specifically, the effectiveness of implemented antislip measures and the overall state of maintenance were thoroughly evaluated. The criteria used for judging building susceptibility during these site visits are illustrated in six example building site surveys provided below:

Building A: Categorized in a ‘medium’ landslide intensity area with a ‘high’ PV. Remarkably, the building shows no significant damage, thus indicative of effective landslide mitigation measures. This observation confirms its high vulnerability rating despite being situated in a moderate-risk area.

Building B: Possessing a ‘medium’ PV and located in an area of ‘medium’ landslide intensity, this building appears stable with no visible damage, aligning with its medium vulnerability assessment.

Building C: Positioned in a high-risk landslide area with a very high PV. Significant perimeter structural damage noted during the survey confirms the model’s assignment and underscores the high-risk status of this building.

Building D: This building, with a ‘medium’ PV in a ‘medium’ landslide intensity zone, is surrounded by terraced reinforced slopes and is well maintained, thus mirroring its moderate vulnerability status consistent with its classification.

Structure E: Assigned a ‘very high’ landslide hazard risk with a corresponding very high PV. Overwhelming structural damage observed and documented in photographs confirms this categorization, thus highlighting the severe vulnerability of the building.

Building F: Classified in a ‘moderate’ landslide hazard risk area with a matching moderate PV. Visible moderate protection measures and the building’s overall condition corroborate this categorization, thus indicating alignment with the expected level of risk.

The findings from the field survey are comprehensively synthesized and presented in Table 4, while Figure 11 visually illustrates the conditions of the buildings relative to their assigned landslide intensity scenarios. The clear alignment between the observed field conditions and the predicted susceptibility categories validates the reliability of the model, thus confirming its effectiveness in accurately assessing landslide risk.

Concurrently, survey forms were completed, and hand-drawn profiles of the selected buildings were created to more effectively verify their vulnerability in the field. These profiles are depicted in Table 5.

Following a comprehensive building vulnerability assessment of Xinxing County, a total of 236 buildings were selected for on-site validation to confirm the accuracy and reliability of the assessment results. These buildings were chosen as representative samples spanning various landslide intensity zones and vulnerability classes, thus ensuring the comprehensiveness and representativeness of the assessment. Details of the sampling and validation results are presented in Table 6.

## 5. Discussion

In refining the research methodology, this study builds upon the concepts of previous researchers to assess building vulnerability. A key distinction from prior methodologies is the enhancement of the definition of landslide intensity. Previously determined by the speed and volume of landslides, it has been expanded to encompass the broader concept of landslide disaster hazard. This enhancement includes a comprehensive evaluation of disaster-causing factors such as topography and geomorphology (elevation, slope gradient, and slope shape), rock and soil body types, geological formations, vegetation cover, watersheds, and land-use types. This holistic approach allows for a more accurate reflection of the potential hazards of landslides, thus recognizing that the risk of occurrence is influenced not only by physical but also by environmental factors. For instance, landslides of identical volume and velocity may present vastly different hazards depending on varying topographic features or geotechnical body types.

This paper presents a large-scale study of building susceptibility under the influence of landslides, thus addressing a gap in the previous research, which has generally not focused on large-scale susceptibility studies. Given the extensive number of buildings assessed, the results are presented through subvillage statistics. This detailed level of assessment permits a nuanced understanding of the specific needs and vulnerabilities of each area. In this study, the building vulnerability index for each village was calculated by aggregating the vulnerability indices of the individual buildings. Additionally, multiple factors were considered for weighting to refine the overall building vulnerability index of the villages.

## 6. Conclusions

This study presents a comprehensive assessment of building vulnerability to landslides on a large scale by integrating quantitative and machine learning methods using Xinxing County in Guangdong Province as a case study. Initially, we collected extensive data on various factors influencing landslide occurrence, including topography, geological structure, vegetation cover, and land use. Using statistical methods, we identified key factors significantly impacting landslide risk. Subsequently, we employed a Convolutional Neural Network (CNN) model to evaluate landslide hazard risks and combined this with an assessment of building resistance based on structural and environmental parameters. This approach allowed us to calculate a detailed vulnerability value (PV) for each building.

The model can accurately and reliably predict the building vulnerability class in most cases, thus achieving an accuracy rate of 94.5% based on on-site validation. Specifically, the landslide hazard assessment categorized the county into four hazard zones—low, medium, high, and extremely high—thus covering areas of approximately 469.41 km², 444.44 km², 396.16 km², and 199.36 km², respectively. The vulnerability assessment identified that 18% of buildings are in low-risk zones, 7% in medium-risk zones, 17% in high-risk zones, and 24% in very high-risk zones. In terms of building vulnerability, the study found that 31.87% of buildings exhibited low vulnerability, 18.00% exhibited medium vulnerability, 34.23% exhibited high vulnerability, and 15.90% exhibited very high vulnerability.

By integrating machine learning with quantitative methods, this study provides a robust tool for disaster risk management, thus enhancing the ability to predict and respond to landslide hazards with greater accuracy. This approach supports the development of targeted mitigation strategies and contributes to the broader field of geohazard management with a scalable and adaptable assessment model.

## Figures and Tables

**Figure 1 sensors-24-04366-f001:**
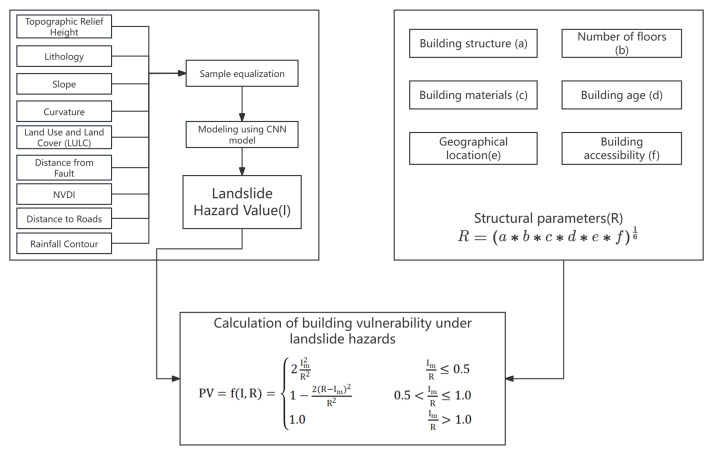
Flowchart showing the methodology for assessing the physical vulnerability of buildings exposed to landslides.

**Figure 2 sensors-24-04366-f002:**
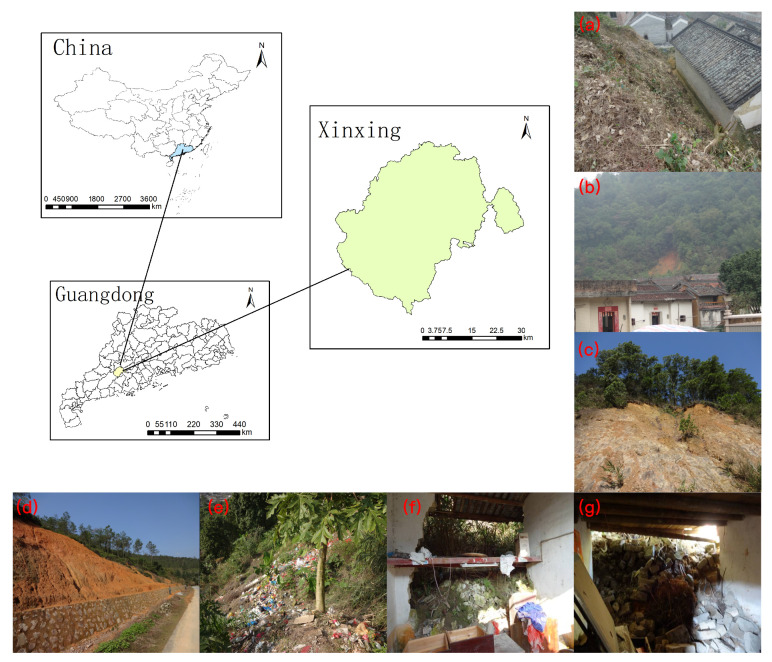
Location map of the study area in Xinxing County and examples of photos of landslide disaster sites (**a**–**g**).

**Figure 3 sensors-24-04366-f003:**
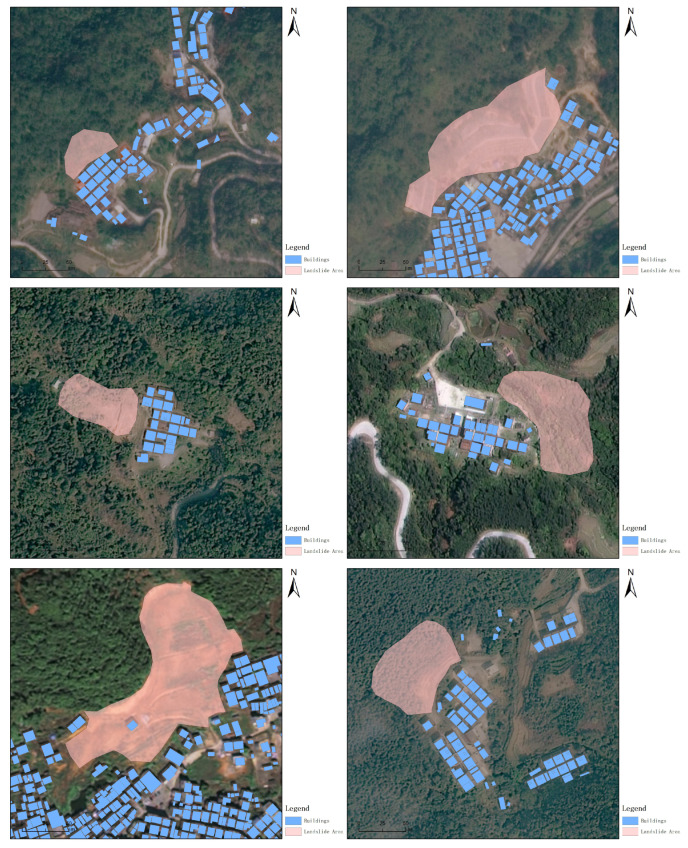
Landslide slope map.

**Figure 4 sensors-24-04366-f004:**
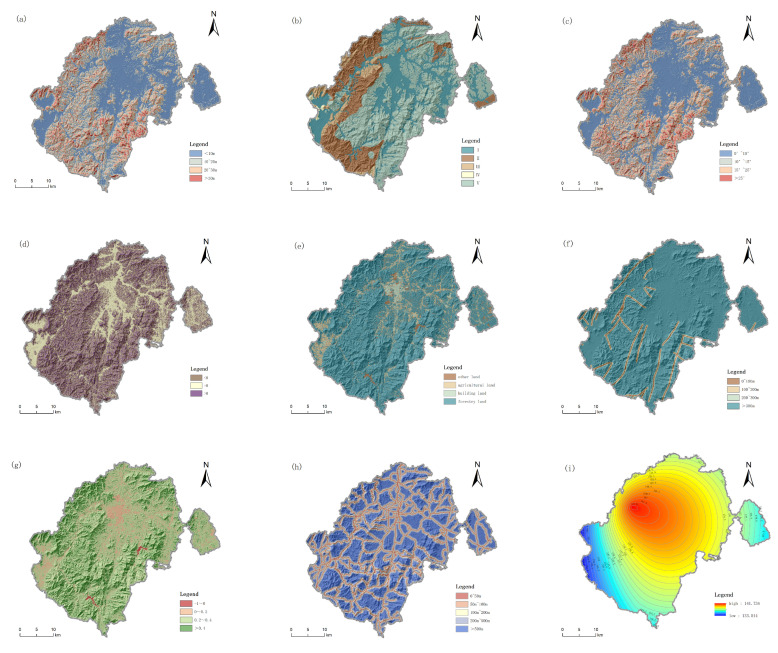
Characteristics of the spatial distribution of the influencing factors. (**a**) Topographic Relief Height, (**b**) Lithology, (**c**) Slope, (**d**) Curvature, (**e**) Land Use and Land Cover (LULC), (**f**) Distance from Fault, (**g**) NVDI, (**h**) Distance to Roads, (**i**) Rainfall Contour.

**Figure 5 sensors-24-04366-f005:**
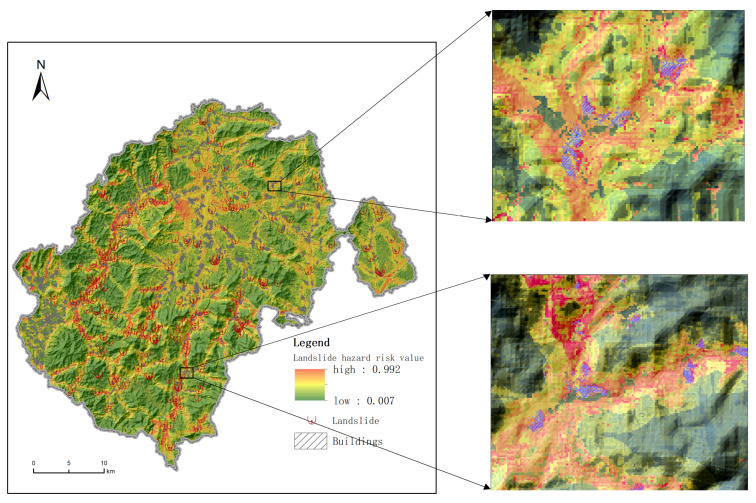
Landslide hazard distribution map of Xinxing County.

**Figure 6 sensors-24-04366-f006:**
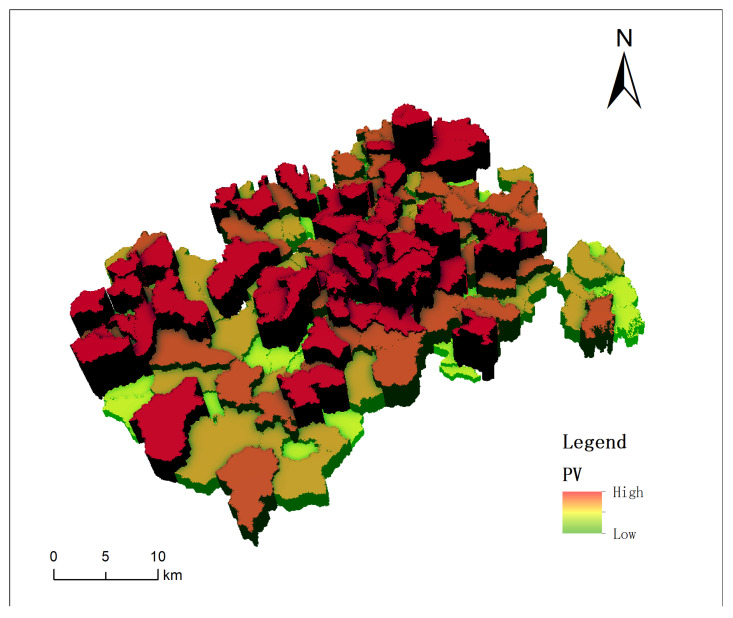
Distribution of building vulnerability values by village.

**Figure 7 sensors-24-04366-f007:**
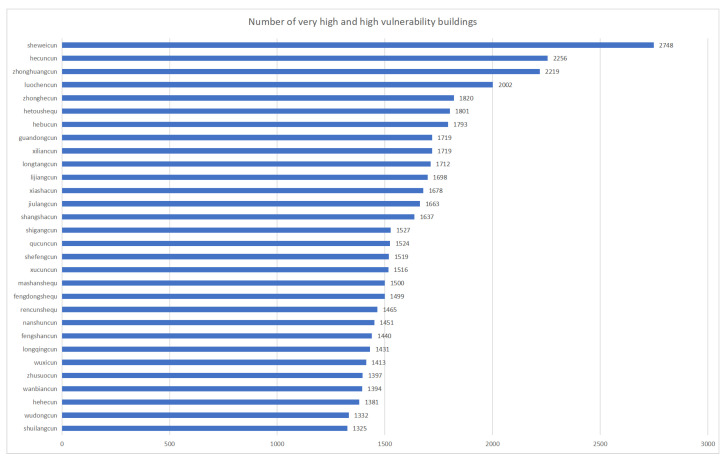
Statistical chart of the number of buildings with very high vulnerability and high vulnerability.

**Figure 8 sensors-24-04366-f008:**
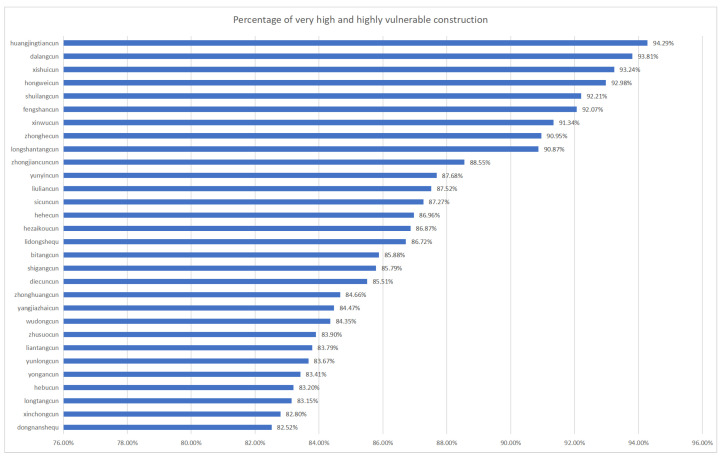
Statistical chart of the percentage of very high vulnerability and high vulnerability buildings.

**Figure 9 sensors-24-04366-f009:**
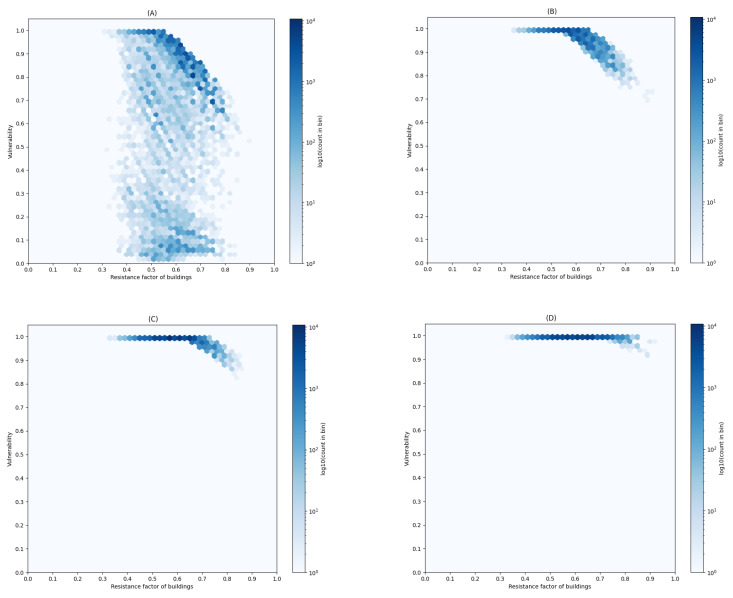
Distributability of building vulnerability for different landslide hazards. (**A**) Low landslide intensity, (**B**) Medium landslide intensity, (**C**) High landslide intensity, (**D**) Very high landslide intensity.

**Figure 10 sensors-24-04366-f010:**
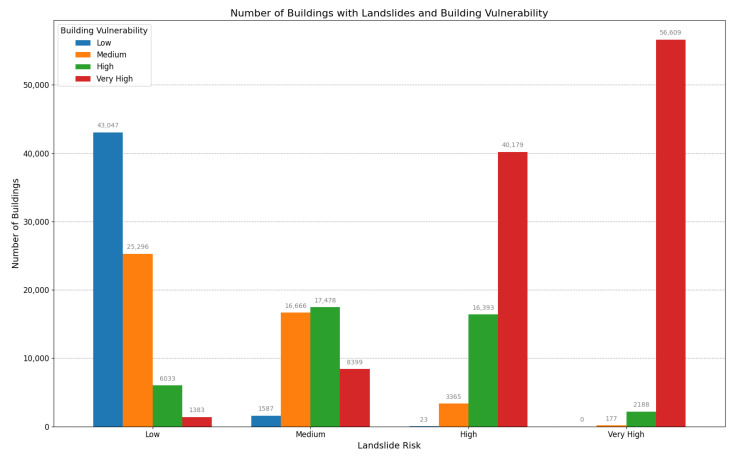
Statistical map of the distribution of the number of buildings at different landslide hazards.

**Figure 11 sensors-24-04366-f011:**
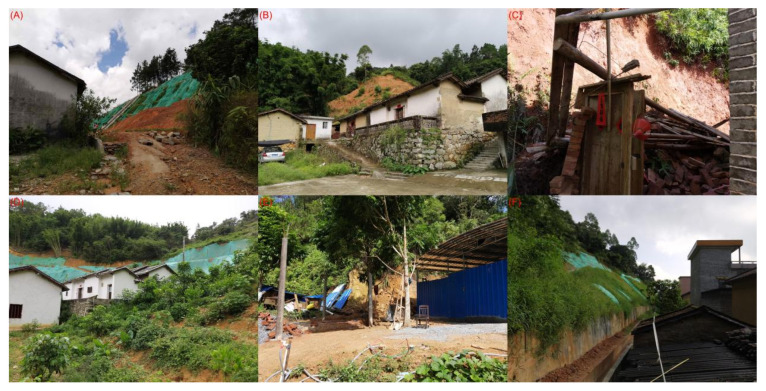
Six different buildings with different physical vulnerability (PV) values under different landslide intensities.(**A**) Medium landslide intensity, high vulnerability, (**B**) Medium landslide intensity, medium vulnerability, (**C**) High landslide intensity, very high vulnerability, (**D**) Medium landslide intensity, medium vulnerability, (**E**) Very high landslide intensity, very high vulnerability, (**F**) Medium landslide intensity, medium vulnerability.

**Table 1 sensors-24-04366-t001:** Resistance coefficients for structural parameters of buildings in Xinxing County.

Structural Parameter	Category	Resistance Factor
Structure Type (sty)	Other Structures	0.25
	Brick and Wood Structure	0.5
	Mixed Structure	0.75
	Reinforced Concrete Structure	1
Number of Floors (st)	One Floor	0.25
	Two Floors	0.5
	Three Floors	0.75
	More than Three Floors	1
Terrain Undulation	0∼14 m	1
	14∼22 m	0.75
	22∼30 m	0.5
	Above 30 m	0.25
Building Accessibility	>500 m	0.2
	225–500 m	0.4
	125–225 m	0.6
	60–125 m	0.8
	0–60 m	1
Rock Factor	Layered Soft Metamorphic Rock Group	0.2
	Layered Strong Karstified Carbonate Rock Group	0.4
	Sand, Gravel, Clay, and Double Soil Layer	0.6
	Layered Hard Clastic Rock Group	0.8
	Blocky Hard-Intrusive Rock Group	1
Building Age	>60 years	0.2
	40–60 years	0.4
	30–40 years	0.6
	15–30 years	0.8
	0–15 years	1

**Table 2 sensors-24-04366-t002:** Xinxing County landslide hazard zoning scale.

Hazard Class	Landslide		Hazard Classification		
	Number	Percentage	Number of Rasters	Area (m^2^)	Percentage
Low-hazard area	24	9.02%	521,562	469.4058	31.10%
Middle-hazard area	82	30.82%	493,817	444.4353	29.44%
High-hazard area	95	35.71%	440,181	396.1629	26.25%
Very high-hazard area	65	24.43%	221,511	199.3599	13.21%
Total	266	1	1,677,071	1509.3639	1

**Table 3 sensors-24-04366-t003:** List of structural attributes of buildings in Wong Hing Tin Village.

ID	Area/m^2^	a	b	c	d	e	f	R	PV
0	105	0.8	0.25	0.75	0.2	0.2	0.8	0.410715	0.835709489
1	90	0.8	0.25	0.75	0.2	0.2	0.8	0.410715	0.835709489
2	85	0.8	0.25	0.75	0.2	0.2	1	0.426277	0.91029835
3	87	0.8	0.25	0.75	0.2	0.2	0.6	0.391487	0.794509013
4	36	0.8	0.25	0.75	0.2	0.2	0.6	0.391487	0.857531181
5	94	0.8	0.25	0.75	0.2	0.2	0.6	0.391487	0.857531181
6	72	0.8	0.25	0.75	0.2	0.2	0.8	0.410715	0.96319616
7	77	0.8	0.25	0.75	0.2	0.2	0.8	0.410715	0.96319616
8	76	0.8	0.25	0.75	0.2	0.2	0.6	0.391487	1
9	72	0.8	0.25	0.75	0.2	0.2	0.6	0.391487	0.784566279
10	71	0.8	0.25	0.75	0.2	0.2	0.6	0.391487	0.784566279
11	33	0.8	0.25	0.75	0.2	0.2	0.8	0.410715	0.933814254
12	21	0.8	0.25	0.75	0.2	0.2	0.8	0.410715	0.882111931
13	98	0.8	0.25	0.75	0.2	0.2	0.8	0.410715	0.991538269
14	36	0.8	0.25	0.75	0.2	0.2	0.8	0.410715	0.768561856
15	51	0.8	0.25	0.75	0.2	0.2	0.8	0.410715	0.488585915
16	79	0.8	0.25	0.75	0.2	0.2	0.6	0.391487	0.857531181
17	35	0.8	0.25	0.75	0.2	0.2	0.6	0.391487	0.735061062

**Table 4 sensors-24-04366-t004:** Classification of observed building damage.

ID	Intensity (I)	PV	Observed Damage	Notes
A	Medium	High	No visible damage	High vulnerability in a moderate risk zone with effective mitigation
B	Medium	Medium	Stable, no overt damage	Moderate vulnerability matching medium landslide intensity
C	High	Very High	Significant peripheral damages	High vulnerability with very high PV value, confirming high risk
D	Medium	Medium	Well-maintained, terraced slopes	Moderate vulnerability with signs of effective stabilization
E	Very High	Very High	Overwhelming structural damage	Severe vulnerability due to very high landslide risk
F	Medium	Medium	Moderate protective measures visible	Medium vulnerability corresponding to medium landslide risk

**Table 5 sensors-24-04366-t005:** Building damage validation form.

**Map name**	Dajiang Town
**Map number**	F49E010017
**Unified number**	445321200037
**Field number**	GC2-011
**Name**	Observation point of Goose Stone Village Ridong Town
**Geographic location**	Yunfu City, Guangdong Province, Xinxing County, Ridong Town, Goose Stone Low Village, Goose Stone Elementary School
**Coordinates**	**Longitude** 112°10′8.27″
	**X coordinate** 2480802
	**Latitude** 22°25′11.37″
	**Y coordinate** 19620360
**Building number**	023
**Building structure type**	Brick structure
**Building name**	Goose Rock Elementary School
**Point Description**	The building point is in a low mountainous terrain with steeply sloping terrain at the point. The stratum of the area is Jurassic Late Jurassic coarse-medium grained porphyritic black mica granite, the weathering degree is strong weathering, and the overlaying layer is residual slope accumulation layer, with sandy clay as the main component. The slope is a rocky slope with a slope direction of 270°, a slope gradient of 80°, a foot elevation of 247 m, a slope height of 5 m, a slope length of 5m, a slope width of 51 m, and a straight profile. The cut slope is used for the construction of a house, which is 6 m wide and 11 m high, 1 m away from the foot of the slope. The vegetation at the top of the slope is well developed, and the vegetation on the slope surface is poorly developed, with a coverage of 50%.
**Remarks**	There is a small-scale collapse on the left side of the slope.
**Plans and sections**	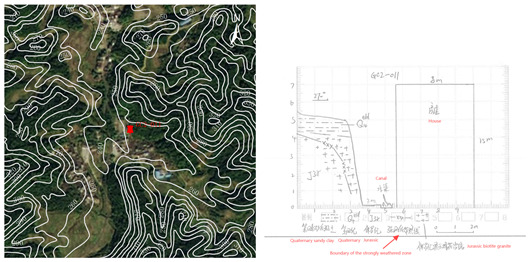

**Table 6 sensors-24-04366-t006:** Sample verification form for construction vulnerability.

Sampling		Verification			
		Low Vulnerability	Medium Vulnerability	High Vulnerability	Very High Vulnerability
Low Vulnerability	60	58	2		
Medium Vulnerability	70		68	2	
High Vulnerability	50			45	5
Very High Vulnerability	56			4	52

## Data Availability

The scripts and datasets used and/or analyzed during the current study are available from the corresponding author on reasonable request.

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
