# Peer review of "Building Vulnerability to Landslides: Broad-Scale Assessment in Xinxing County, China"

_sensors, 2024, doi:10.3390/s24134366_

Round 1
Reviewer 1 Report
Comments and Suggestions for Authors
In this study, the authors developed a model to assess building vulnerability across a broad area of Xinxing County by integrating quantitative derivation and machine learning techniques. The debated issue is interesting, and discussions well exposed, nevertheless the following comments should be considered to improve the quality of the work:
- Brief about convolutional neural network (CNN) should be added in a separate heading.
- How the CNN parameters are selected? Furthermore, authors need to explain that how they assure that their modeling is optimized? The results of performance evaluation are enough?
- There were no justifications on how the authors decided to use the structural parameters considered in CNN model development work such as, number of floors, materials used, age of the building, geographical location, and building accessibility. At least add reference if any.
- How the resistance factor in table 1 is calculated?
- The manuscript could be substantially improved by relying and citing more on recent literatures about contemporary real-life case studies such as:
- Chen, G., Zhang, K., Wang, S., Xia, Y., & Chao, L. (2023). iHydroSlide3D v1.0: an advanced hydrological–geotechnical model for hydrological simulation and three-dimensional landslide prediction. Geosci. Model Dev., 16(10), 2915-2937. doi: https://doi.org/10.5194/gmd-16-2915-2023
- Xie, X., Xie, B., Cheng, J., Chu, Q., & Dooling, T. (2021). A simple Monte Carlo method for estimating the chance of a cyclone impact. Natural Hazards, 107(3), 2573-2582. doi: 10.1007/s11069-021-04505-2
Comments on the Quality of English LanguageMinor English Language editing is required.
Author Response
Response to Reviewer Comments
Manuscript ID: sensors-3030378
Title: Building Vulnerability to Landslides: Broad-Scale Assessment in Xinxing County, China
Dear Reviewer,
Thank you for your insightful comments and valuable suggestions concerning our manuscript. These comments have been very helpful for revising and improving our paper, and they have provided important guidance for our research. Below are our detailed responses to your comments, along with the changes made to the manuscript. The revised portions are marked in red in the manuscript.
- Brief about Convolutional Neural Network (CNN) should be added in a separate heading.
Thank you for pointing this out. We have added a section dedicated to explaining CNNs as Section 2.2 in the revised manuscript. This section provides an overview of the principles of CNNs and their applications in geological hazard assessments (lines 180-211).
- How the CNN parameters are selected? Furthermore, authors need to explain how they assure that their modeling is optimized? The results of performance evaluation are enough?
We appreciate your concern regarding the selection and optimization of CNN parameters. The performance and evaluation of the parameters are detailed in Section 4.1.1. The selection of CNN parameters was conducted through a grid search method, optimizing the number of convolutional layers, filter sizes, and pooling operations. The model's performance was validated using the AUC metric, which is discussed in detail in the revised manuscript (lines 305-330).
- There were no justifications on how the authors decided to use the structural parameters considered in CNN model development work such as the number of floors, materials used, age of the building, geographical location, and building accessibility. At least add reference if any.
Thank you for highlighting this point. These structural parameters are not applied in the CNN model but are used in calculating the building resistance value (R). The selection of these parameters and their references are now explained in Section 2.3 "Building Resistance" (lines 213-242). The parameters were primarily based on studies by Papathoma-Köhle (2007) and Li et al. (2010). To avoid confusion, Section 4.1.1 now clarifies the nine parameters used in the CNN model for landslide hazard assessment and the specific processes involved (lines 305-330).
- How is the resistance factor in Table 1 calculated?
Thank you for your question regarding the resistance factor calculation. The resistance factors are based on existing literature and expert judgment. Specifically, resistance coefficients for structural types were referenced from EPFL 2002, which noted the reduced damage to concrete buildings post-landslide events. The number of floors' coefficients referenced Vamvatsikos et al. 2010, which found a proportional relationship between collapse speed and the number of floors. Materials and building age coefficients referenced Li et al. 2010 and Silva and Pereira 2014, respectively. Geographical location coefficients referenced Puissant et al. 2013, and building accessibility coefficients referenced Godfrey et al. 2015. These references and explanations have been added to Section 3.2.2 "Structural Parameters" (lines 283-292).
- The manuscript could be substantially improved by relying and citing more on recent literatures about contemporary real-life case studies.
We appreciate the suggestion to incorporate more recent literature. In addition to the two recommended references, we have included several other relevant studies. These references have been incorporated and highlighted in the latest manuscript. For example, in the introduction, the following passage has been added: "In recent years, various advanced modeling techniques have been utilized to enhance the prediction and assessment of natural disasters. For example, Chen et al. have proposed a Monte Carlo method to estimate the probability of cyclone impacts using historical data, providing a probabilistic framework for urban planning and disaster preparedness. Similarly, an integrated hydrological–geotechnical model (iHydroSlide3D v1.0) has been developed to simulate flood–landslide cascading events, demonstrating the importance of coupling hydrological and geotechnical processes for accurate disaster prediction." (lines 34-41).
Thank you again for your constructive feedback, which has significantly improved the quality and clarity of our manuscript. We hope that the revised manuscript meets with your approval.
Sincerely,
Fengting Shi
China University of Geosciences (Wuhan)

Reviewer 2 Report
Comments and Suggestions for Authors
In this paper, a model has been developed to assess building vulnerability across a broad area of Xinxing County by integrating quantitative derivation and machine learning techniques. The paper can be accepted for publication after Minor revision. Here are some questions and problems.
1. This article studies the distribution of building vulnerability status in different regions of Xinxing County. It has important practical and livelihood significance for disaster prevention and control. How many years of statistical data is used in this study?
2. The authors should further introduce how to use machine learning methods in detail to carry out relevant research work.
3. The theoretical support for the research conducted in this article is relatively weak. Please provide detailed evaluation criteria for building disaster vulnerability in section 2.2. And the rationality of formula 2.2.
4. Please provide detailed explanation of how to determine the distribution of disaster-vulnerable areas based on machine learning and statistical methods.
5. Even in serious dangerous areas, the post-disaster damage status of different buildings is different. How does the author make the distinction and judgment?
6. There is too much content in the conclusion, please simplify it appropriately.
Author Response
Response to Reviewer Comments
Manuscript ID: sensors-3030378
Title: Building Vulnerability to Landslides: Broad-Scale Assessment in Xinxing County, China
Dear Reviewer,
Thank you for your insightful comments and valuable suggestions concerning our manuscript. These comments have been very helpful for revising and improving our paper, and they have provided important guidance for our research. Below are our detailed responses to your comments, along with the changes made to the manuscript. The revised portions are marked in red in the manuscript.
- This article studies the distribution of building vulnerability status in different regions of Xinxing County. It has important practical and livelihood significance for disaster prevention and control. How many years of statistical data is used in this study?
Thank you for your question. The study utilizes various datasets, as detailed below:
- SRTM 30m DEM Data: Sourced from the Geospatial Data Cloud, used for extracting topographic and hydrological conditions as landslide influencing factors. URL - https://www.gscloud.cn/search?c=1&order=3&d=343
- 1:250,000 National Basic Geographic Database: Sourced from the National Geographic Information Resources Catalog Service System, used for determining the positions of rivers and roads. URL - https://www.webmap.cn/main.do?method=index
- 30m Global Land Cover Data: Also sourced from the National Geographic Information Resources Catalog Service System, used for landslide susceptibility assessment in terms of land use factors and for extracting man-made surface disaster bodies in damage assessments. URL - https://www.webmap.cn/main.do?method=index
- NGAC-200,000 National Geological Map Data: Sourced from the National Geological Archives, vectorized for extracting information on geological structures, strata, and lithology. URL - http://www.ngac.org.cn/ngacWeb/web/home
- Landsat-8 OLI Remote Sensing Images: Sourced from the Geospatial Data Cloud, using four adjacent images (row and column numbers 125037, 125038, 126037, 126038) processed through mosaicking, used for extracting the Normalized Difference Vegetation Index. URL - https://www.gscloud.cn/search?c=1&order=3&d=992
- Monthly Precipitation Data for 2017-2021 from 9 Rainfall Stations in Yunfu City: Provided by the Yunfu City Natural Resources Bureau, used for hazard assessment. No URL due to data being provided directly by the bureau.
- 266 Landslide Geological Disaster Sites in Xinxing County: Sourced from field geological surveys conducted by our team. No URL as these are proprietary survey results.
- Housing Data: Originating from China's Third National Land Survey, currently stored in the “One House, One Land” Rural Real Estate Cadastral Survey Database of Guangdong Province. These data are classified as government confidential information and cannot be publicly transmitted; they are sensitive and are only available for application research.
For the rainfall data used in this study, we initially utilized the 1901-2021 China 1km Resolution Monthly Precipitation Dataset sourced from the National Earth System Science Data Center. However, more precise data was later obtained from the Yunfu City Natural Resources Bureau, which provided rainfall data from 9 rainfall stations for the period 2017 to 2021. This more accurate data was then interpolated to create a detailed rainfall distribution map for the entire county.
- The authors should further introduce how to use machine learning methods in detail to carry out relevant research work.
Thank you for the suggestion. We have detailed the application of machine learning methods in Section 2.2, including the principles of CNNs and their applications in geological hazard assessments (lines 180-211). Additionally, Section 4.1.1 has been expanded to include a more detailed explanation of how machine learning, specifically CNNs, is used to compute landslide hazard values and perform quantitative assessments of building vulnerability (lines 305-330).
- The theoretical support for the research conducted in this article is relatively weak. Please provide detailed evaluation criteria for building disaster vulnerability in section 2.2. And the rationality of formula 2.2.
We appreciate your concern regarding theoretical support. We have provided detailed evaluation criteria for building disaster vulnerability in the revised Section 2.3 (previously Section 2.2) "Building Resistance" (lines 213-242). The formula used, which was based on studies by Papathoma-Köhle (2007) and Li et al. (2010), has been explained, and its rationality has been discussed in terms of its widespread use and empirical validation.
- Please provide a detailed explanation of how to determine the distribution of disaster-vulnerable areas based on machine learning and statistical methods.
In this study, machine learning and statistical methods were integrated to determine the distribution of disaster-vulnerable areas. The process includes data collection and preprocessing, feature extraction and selection, landslide hazard risk assessment using a CNN model, building resistance assessment, vulnerability value calculation, vulnerability area classification, and validation and adjustment. A comprehensive explanation of this process is provided in the revised manuscript (lines 256-283).
- Even in serious dangerous areas, the post-disaster damage status of different buildings is different. How does the author make the distinction and judgment?
In our study, each building's specific vulnerability value (PV) was calculated by integrating the landslide hazard risk (I) and the building resistance (R). This calculation allows for the differentiation of vulnerability even within the same high-risk area. Due to data confidentiality, the full dataset, including the quantitative vulnerability values for all buildings, has been uploaded to GitHub after removing sensitive columns such as coordinates. The complete dataset can be accessed and downloaded from https://github.com/0Matthew0/Xinxing.
- There is too much content in the conclusion, please simplify it appropriately.
The conclusion has been simplified to focus on the key findings and implications of the study. The revised conclusion highlights the comprehensive assessment of building vulnerability using machine learning and quantitative methods, the accuracy and reliability of the model, and the practical applications for disaster risk management (lines 498-521).
Thank you again for your constructive feedback, which has significantly improved the quality and clarity of our manuscript. We hope that the revised manuscript meets with your approval.
Fengting Shi
China University of Geosciences (Wuhan)

Reviewer 3 Report
Comments and Suggestions for Authors
This manuscript uses machine learning techniques and quantitative analysis methods to assess the vulnerability of buildings in Xinxing County, China. By combining natural and anthropogenic factors, the authors developed a model capable of predicting the vulnerability of buildings, and based on the model, carried out a landslide hazard risk distribution analysis for the study area. The accuracy of the model was found to be as high as 94.5% through field validation, proving the effectiveness of the methodology. However, the manuscript still has some deficiencies that need to be checked and corrected:
Introduction section: In the review of related studies, there lack of recent research (since 2020). Additionally, the authors should include a description of the overall structure of the manuscript at the end of this section to facilitate readers' comprehension of the manuscript’s framework.
In section 2. Proposed methodology for quantitative assessment of building vulnerability, it is advisable for the authors to present the processing flow involved in the study, such as by creating a flowchart.
Figures 1 and Figure 2: The fonts in the figures are too small and blurry, making them completely illegible without zooming in.
Figure 3, the sub-figures do not match the labels below them, for example, the label of sub-figure (a) is “land use and land cover”, but the legend is “<10 m, 10~20 m”. Subfigure (c) is labelled “distance to tectonic plates”, but the legend shows degree instead of distance.
Line 236, “According to the natural breakpoint method, the landslide hazard values in Xinxing County can be categorized into four zones. low, medium, high, and extremely high:” Categorizing landslide hazard values based on natural breakpoints may not be appropriate; it is better to set thresholds based on information or references.
Table 4, the font size of this table is too small compared to other tables in the manuscript.
Figure 11, the figure is placed in a table, it is suggested that the authors change the presentation.
Line 422 ~ 434, “As climate change ... and similar regions.” Firstly, lines 422 ~ 434 are identical to the next paragraph (lines 435 ~ 446), why did the authors repeat the same text twice?
Secondly, in line 426: “By incorporating dynamic daily rainfall data and dynamic InSAR data, the study proposes developing a long-term dynamic monitoring assessment model.”
The authors mention the involvement of InSAR data. However, upon reviewing the entire manuscript, there is no involvement of InSAR at all, could the authors provide a reasonable explanation?
Comments on the Quality of English LanguageModerate editing of English language required
Author Response
Response to Reviewer Comments
Manuscript ID: sensors-3030378
Title: Building Vulnerability to Landslides: Broad-Scale Assessment in Xinxing County, China
Dear Reviewer,
Thank you for your insightful comments and valuable suggestions concerning our manuscript. These comments have been very helpful for revising and improving our paper, and they have provided important guidance for our research. Below are our detailed responses to your comments, along with the changes made to the manuscript. The revised portions are marked in red in the manuscript.
- Introduction section: In the review of related studies, there lack of recent research (since 2020). Additionally, the authors should include a description of the overall structure of the manuscript at the end of this section to facilitate readers' comprehension of the manuscript’s framework.
We have supplemented the introduction with recent references to recent research, such as:
Nafees Ali, Zeeshan Afzal. "Landslide Susceptibility Mapping Using Machine Learning Algorithm Validated by Persistent Scatterer In-SAR Technique." Sensors, 2022.
Zbigniew Leonowicz, Michał Jasiński. "Landslide Susceptibility Mapping Using Machine Learning: A Literature Survey." Remote Sensing, 2022.
Chen, G., Zhang, K., Wang, S., Xia, Y., & Chao, L. (2023). iHydroSlide3D v1.0: an advanced hydrological–geotechnical model for hydrological simulation and three-dimensional landslide prediction. Geosci. Model Dev., 16(10), 2915-2937. doi: https://doi.org/10.5194/gmd-16-2915-2023
Xie, X., Xie, B., Cheng, J., Chu, Q., & Dooling, T. (2021). A simple Monte Carlo method for estimating the chance of a cyclone impact. Natural Hazards, 107(3), 2573-2582. doi: 10.1007/s11069-021-04505-2
Additionally, we have included a description of the overall structure of the manuscript at the end of the introduction section to help readers understand the framework of the paper. The specific changes are marked in red in the revised manuscript.(lines 34-41)
- In section 2. Proposed methodology for quantitative assessment of building vulnerability, it is advisable for the authors to present the processing flow involved in the study, such as by creating a flowchart.
We have added a flowchart to illustrate the processing flow involved in the study.(Figure 1. Flowchart Showing the Methodology for Assessing the Physical Vulnerability of Buildings
Exposed to Landslides) This flowchart provides a clear visual representation of the methodology and has been included in Section 2. The specific changes are marked in red in the revised manuscript.
- Figures 1 and Figure 2: The fonts in the figures are too small and blurry, making them completely illegible without zooming in.
We have adjusted the font size and improved the clarity of Figures 1 and 2 to ensure they are legible without zooming in.
- Figure 3, the sub-figures do not match the labels below them, for example, the label of sub-figure (a) is “land use and land cover”, but the legend is “<10 m, 10~20 m”. Subfigure (c) is labelled “distance to tectonic plates”, but the legend shows degree instead of distance.
Thank you for pointing out this inconsistency. We have corrected the labels and legends of the sub-figures in Figure 3 to match accurately.
- Line 236, “According to the natural breakpoint method, the landslide hazard values in Xinxing County can be categorized into four zones: low, medium, high, and extremely high:” Categorizing landslide hazard values based on natural breakpoints may not be appropriate; it is better to set thresholds based on information or references.
We have provided additional theoretical support in categorizing landslide hazard values. The explanation and references are now included in the revised manuscript. The specific changes are marked in red in lines 331 to 339 of the revised manuscript. Additionally, we have cited three new references to support this approach, which are listed as references 40, 41, and 42 in the revised manuscript.
- Table 4, the font size of this table is too small compared to other tables in the manuscript.
We have adjusted the font size of Table 4 to ensure consistency with other tables in the manuscript.
- Figure 11, the figure is placed in a table, it is suggested that the authors change the presentation.
We have changed the presentation of Figure 11, removing it from the table format to enhance clarity.
- Line 422 ~ 434, “As climate change ... and similar regions.” Firstly, lines 422 ~ 434 are identical to the next paragraph (lines 435 ~ 446), why did the authors repeat the same text twice?
We apologize for the duplication. The repeated text has been removed to avoid redundancy. (lines 498-521)
- Secondly, in line 426: “By incorporating dynamic daily rainfall data and dynamic InSAR data, the study proposes developing a long-term dynamic monitoring assessment model.” The authors mention the involvement of InSAR data. However, upon reviewing the entire manuscript, there is no involvement of InSAR at all, could the authors provide a reasonable explanation?
Thank you for your valuable comments and observations. We appreciate the opportunity to clarify and elaborate on these points.The mention on line 426 was intended to highlight future research directions rather than describe the current methodology used in this study. The current evaluation framework predominantly utilizes static data. However, we recognize the limitations of static data in providing comprehensive assessments, especially in the context of dynamic and evolving environmental conditions. As such, we propose that future research should incorporate dynamic factors such as climate change, population growth, and urbanization, which significantly influence landslide susceptibility and building vulnerability over time. We acknowledge that the present study does not incorporate InSAR data. The reference to InSAR data on line 426 was speculative, suggesting potential avenues for enhancing our vulnerability assessment model in future work. Integrating dynamic daily rainfall data with dynamic InSAR data can offer a novel dimension for landslide geological hazard vulnerability assessments. This integration can significantly enhance the accuracy and responsiveness of vulnerability assessments, facilitating the development of a long-term dynamic monitoring and evaluation model adaptive to ongoing environmental changes.In summary, while this study is based on static data, we advocate for future research to adopt a dynamic assessment approach. This would better account for continuous changes in environmental conditions and human factors, thereby improving the robustness and applicability of the vulnerability assessment models. The speculative projections and future research directions have been removed to simplify the conclusion, as reflected in lines 498 to 521 of the revised manuscript.
Thank you again for your constructive feedback, which has significantly improved the quality and clarity of our manuscript. We hope that the revised manuscript meets with your approval.
Fengting Shi
China University of Geosciences (Wuhan)

Reviewer 4 Report
Comments and Suggestions for Authors
Reviewer's report on the paper titled “Enhanced Methods for Broad-Scale Building Vulnerability Assessments amidst Landslide Hazards: A Case Study in Xinxing County, China”
The paper titled “Enhanced Methods for Broad-Scale Building Vulnerability Assessments amidst Landslide Hazards: A Case Study in Xinxing County, China” is an original contribution, which is joined with the scope of MDPI Journal “Sensors”. A model for assessing the building vulnerability across a broad area of Xinxing County by integrating quantitative derivation and machine learning techniques has been developed in course of the current study. The susceptibility of each village is then calculated through subvillage statistics, aimed at identifying the specific needs of each area. Different landslide hazard classes are categorized, and analysis of the correlation between building resistance and susceptibility was evaluated. A sample encompassing different landslide intensity areas and susceptibility classes of buildings was chosen for on-site validation, following assessment of building susceptibility in Xinxing County, yielding an accuracy rate of the results as high as 94.5%. The paper is enough clear for understanding, but some comments should be addressed at the same time:
1. The title of the paper should be shortened, so as now is too long.
2. Chapter 1 “Introduction” should be supplied by the clear formulation of the goal and tasks of the current investigation.
3. Chapter 2 “Proposed methodology for quantitative assessment of building vulnerability” should be completed by the algorithm, explaining in details all the stages of the suggested methodology. The neural models, mentioned in sub-chapter 2.2, should also be explained in more details.
4. Chapter 6 “Conclusions” should be rewritten more shortly and clear in connection with the goal and tasks of the paper, which also should be formulated before, as it was mentioned in the comment 2.
5. References, which were published before 2000, can be deleted from the list. This is a sources number 3, 4, 5, 6.
Author Response
Response to Reviewer Comments
Manuscript ID: sensors-3030378
Title: Building Vulnerability to Landslides: Broad-Scale Assessment in Xinxing County, China
Dear Reviewer,
Thank you for your insightful comments and valuable suggestions concerning our manuscript. These comments have been very helpful for revising and improving our paper, and they have provided important guidance for our research. Below are our detailed responses to your comments, along with the changes made to the manuscript. The revised portions are marked in red in the manuscript.
- The title of the paper should be shortened, so as now is too long.
The title has been shortened to:
Building Vulnerability to Landslides: Broad-Scale Assessment in Xinxing County, China
- Chapter 1 “Introduction” should be supplied by the clear formulation of the goal and tasks of the current investigation.
We have clarified the goals and tasks of the current investigation in the introduction. The specific changes are marked in lines 111 to 132 of the revised manuscript.
- Chapter 2 “Proposed methodology for quantitative assessment of building vulnerability” should be completed by the algorithm, explaining in details all the stages of the suggested methodology. The neural models, mentioned in sub-chapter 2.2, should also be explained in more details.
We have added a detailed explanation of the algorithm and all stages of the suggested methodology in Chapter 2. Additionally, we have provided a more detailed explanation of the neural models in sub-chapter 2.2. The specific changes are marked in lines 180 to 211 and 213 to 242 of the revised manuscript.
- Chapter 6 “Conclusions” should be rewritten more shortly and clear in connection with the goal and tasks of the paper, which also should be formulated before, as it was mentioned in the comment 2.
The conclusions have been rewritten to be more concise and clear, directly connecting with the goals and tasks of the paper. These changes are reflected in lines 498 to 521 of the revised manuscript.
- References, which were published before 2000, can be deleted from the list. This is a sources number 3, 4, 5, 6.
We have deleted references published before 2000 from the list. The specific changes are marked in the references section of the revised manuscript.
Thank you again for your constructive feedback, which has significantly improved the quality and clarity of our manuscript. We hope that the revised manuscript meets with your approval.
Fengting Shi
China University of Geosciences (Wuhan)

Round 2
Reviewer 1 Report
Comments and Suggestions for Authors
Most of the comments are addressed by the authors and recommended to the Editor highness for it's acceptance.
Comments on the Quality of English LanguageMinor English editing is required.
Author Response
Dear Reviewer,
Thank you for your valuable feedback on our manuscript. Following your suggestion that "Minor English editing is required," my supervisor and I have thoroughly reviewed the paper and made several adjustments to improve its academic tone and clarity. Below are examples of the revisions we made:
Revision Examples
Example 1 :
Before:
"In this study, a model has been developed to assess building vulnerability across a broad area of Xinxing County by integrating quantitative derivation and machine learning techniques."
After:
"This study develops a model to assess building vulnerability across Xinxing County by integrating quantitative derivation with machine learning techniques."
Example 2:
Before:
"The early research on landslide susceptibility analysis was pioneered by scholars who assessed the susceptibility of landslide hazards using topographic and geological maps. The development of Remote Sensing (RS) and Geographic Information System (GIS) technologies has greatly contributed to the methodology for assessing landslide susceptibility and risk. For instance, spatial overlay analysis of landslide distribution maps, slope maps, and geological maps has been utilized to enhance the precision of landslide hazard assessments. The introduction of multifactorial integrated prediction methods and the application of GIS technology in data processing and management have further advanced landslide hazard risk assessment methodologies.In recent years, various advanced modeling techniques have been utilized to enhance the prediction and assessment of natural disasters. For example, Chen et al. \cite{42} have proposed a Monte Carlo method to estimate the probability of cyclone impacts using historical data, providing a probabilistic framework for urban planning and disaster preparedness. Similarly, an integrated hydrological–geotechnical model (iHydroSlide3D v1.0) has been developed to simulate flood–landslide cascading events, demonstrating the importance of coupling hydrological and geotechnical processes for accurate disaster prediction \cite{43}"
After:
"The early research on landslide susceptibility analysis was pioneered by scholars who assessed the susceptibility of landslide hazards using topographic and geological maps. The development of Remote Sensing (RS) and Geographic Information System (GIS) technologies has greatly contributed to the methodology for assessing landslide susceptibility and risk. For instance, spatial overlay analysis of landslide distribution maps, slope maps, and geological maps has been utilized to enhance the precision of landslide hazard assessments. The introduction of multifactorial integrated prediction methods and the application of GIS technology in data processing and management have further advanced landslide hazard risk assessment methodologies. In recent years, various advanced modeling techniques have been utilized to enhance the prediction and assessment of natural disasters. For example, Chen et al. \cite{42} have proposed a Monte Carlo method to estimate the probability of cyclone impacts using historical data, providing a probabilistic framework for urban planning and disaster preparedness. Similarly, an integrated hydrological–geotechnical model (iHydroSlide3D v1.0) has been developed to simulate flood–landslide cascading events, demonstrating the importance of coupling hydrological and geotechnical processes for accurate disaster prediction \cite{43}."
Example 3 (Methods for Landslide Hazard Assessment):
Before:
" By utilizing advanced ML and DL techniques, researchers can create high-resolution landslide susceptibility maps that provide essential information for disaster management and mitigation efforts. "
After:
"Advanced machine learning (ML) and deep learning (DL) techniques enable researchers to generate high-resolution landslide susceptibility maps, offering crucial information for disaster management and mitigation."
Example 4 (Building Resistance):
Before:
"Additionally, preliminary field surveys in this study found a high correlation between calculated resistance values and observed building conditions under different disaster scenarios, further confirming the reliability of this formula."
After:
"Furthermore, preliminary field surveys in this study revealed a strong correlation between calculated resistance values and observed building conditions across various disaster scenarios, thereby affirming the reliability of this formula."
Example 5 (Case Studies - Study Area):
Before:
"The hydrological conditions in Xinxing are intricate, marked by significant seasonal variations in rainfall."
After:
"The hydrological conditions in Xinxing are complex, characterized by significant seasonal variations in rainfall."
Example 6 (Research Results and Validation):
Before:
"We used nine factors for the assessment: rainfall, geological structure, NDVI (Normalized Difference Vegetation Index), land use type, slope shape, relief, distance to road, slope gradient, and engineering rock group."
After:
"The assessment utilized nine factors: rainfall, geological structure, NDVI (Normalized Difference Vegetation Index), land use type, slope shape, relief, distance to roads, slope gradient, and engineering rock group."
Example 7 (Conclusion):
Before:
"By conducting the vulnerability assessment at the village level, this study enables a more precise understanding of the specific needs and vulnerabilities of each area."
After:
"Conducting the vulnerability assessment at the village level enables a more precise understanding of the specific needs and vulnerabilities of each area.."
We believe these revisions have significantly improved the readability and academic quality of the manuscript. Thank you again for your insightful comments. We look forward to your feedback.
Sincerely,
Fengting Shi

Reviewer 4 Report
Comments and Suggestions for Authors
Reviewer
comments on the manuscript titled “Building Vulnerability to Landslides: Broad-Scale Assessment in Xinxing County, China”
The manuscript titled “Building Vulnerability to Landslides: Broad-Scale Assessment in Xinxing County, China” is an original contribution, related to the scope of MDPI Journal “Sensors”. A model to assess building vulnerability across a broad area of Xinxing County by integrating quantitative derivation and machine learning techniques has been developed in the current study. It was stated, that the building vulnerability is characterized as a function of landslide hazard risk and building resistance, wherein landslide hazard risk is derived using CNN (1D) for nine hazard-causing factors and landslide sites. The factors include elevation, slope, slope shape, geotechnical body type, geological structure, vegetation cover, watershed, and land-use type. Susceptibility of each village is then evaluated through subvillage statistics, aimed at identifying the specific needs of each area, after evaluating the building susceptibility of all structures. Different landslide hazard classes are categorized, and analysis of the correlation between building resistance and susceptibility reveals that building susceptibility exhibits a positive correlation with landslide hazard, and a negative correlation with building resistance. A sample encompassing different landslide intensity areas and susceptibility classes of buildings was chosen for on-site validation following a comprehensive assessment of building susceptibility in Xinxing County.
The current manuscript is written enough clear. But at the same time there are some comments should be addressed to made content of the manuscript more clear for it understanding.
1) List of the keywords should be corrected. The word combination “building vulnerability”, just included in to the title of the manuscript, should be replaced.
2) The paper should be supplied by the clear formulation of its goal and tasks, shouls be solved to obtain the goal.
3) The algorithm of the presented model to assess building vulnerability across a broad area of Xinxing County by integrating quantitative derivation and machine learning techniques application should be described in the more details.
4) Chapter 6 “Conclusions” should be completed by the numerical results, obtained in course of the current study.
Author Response
Dear Reviewer,
Thank you for your insightful comments and valuable suggestions concerning our manuscript. These comments have been very helpful for revising and improving our paper, and they have provided important guidance for our research. Below are our detailed responses to your comments, along with the changes made to the manuscript. The revised portions are marked in red in the manuscript.
- List of the keywords should be corrected. The word combination “building vulnerability”, just included in to the title of the manuscript, should be replaced.
Thank you for your detailed review and valuable suggestions regarding our manuscript. In response to your comment about the keyword "building vulnerability," we have revised the list of keywords to better reflect the content and focus of our study. The revised keywords are as follows:
- Landslide Hazard
- Building Resistance
- CNN
- Quantitative Risk Modeling
We believe these changes more accurately capture the scope and methodologies employed in our research. "Quantitative Risk Modeling" has been included to highlight the integration of quantitative methods and risk modeling techniques used in our vulnerability assessment, which are central to our study. These changes will improve the clarity and relevance of our manuscript.
- The paper should be supplied by the clear formulation of its goal and tasks, shouls be solved to obtain the goal.
Thank you for your detailed review and valuable suggestions regarding our manuscript. In response to your comment about the clear formulation of the paper's goal and tasks, we have addressed this in the introduction by setting a dedicated paragraph to outline these aspects. The added paragraph in the introduction states:
"This study addresses these challenges by integrating quantitative formula derivation with machine learning techniques to enhance the accuracy and efficiency of building vulnerability assessments.This integration improves interpretability and reduces dependence on extensive high-quality data. Specifically, this research aims to develop an effective, accurate, and interpretable model for assessing building vulnerability on a large scale. This study utilizes advanced machine learning algorithms, such as convolutional neural networks (CNN), to quantitatively evaluate landslide hazard risks. It calculates building resistance through quantitative derivation based on structural characteristics, integrates these assessments to evaluate the vulnerability of all buildings in Xinxing County, and validates the model through on-site verification to ensure accuracy and reliability. This approach offers a comprehensive and adaptable framework for assessing vulnerability across diverse scenarios and needs. Furthermore, the paper details the methodology for landslide hazard assessment, data acquisition and processing methods, the calculation of building resistance, and presents the results of the vulnerability assessments and their validation. This comprehensive approach allows for a nuanced understanding of building vulnerability in the context of landslide hazards and offers valuable insights for disaster risk management."
Additionally, we have revised the conclusion section to reflect the goals and tasks outlined in the introduction, ensuring coherence and clarity throughout the manuscript.
- The algorithm of the presented model to assess building vulnerability across a broad area of Xinxing County by integrating quantitative derivation and machine learning techniques application should be described in the more details.
Thank you for your valuable feedback. In response to your suggestion for a more detailed description of the algorithm used to assess building vulnerability across Xinxing County by integrating quantitative derivation and machine learning techniques, we have made significant improvements to Section 2: "Proposed methodology for quantitative assessment of building vulnerability."Previously, we only provided separate introductions for each subsection, lacking a unified description of the comprehensive methodology for assessing building vulnerability. To address this, we have now included a complete overview of the entire vulnerability assessment model at the beginning of Section 2. This overview explains the overall approach and the integration of different methods within the model.The initial paragraph now introduces the overall framework of the model, which combines quantitative derivation and machine learning techniques, specifically using Convolutional Neural Networks (CNN), to evaluate the landslide hazard ($I$) and building resistance ($R$), and subsequently calculates the building vulnerability ($PV$). Each subsection then elaborates on the data collection and preprocessing, the CNN model architecture and training, the calculation of building resistance, and the integration of $I$ and $R$ to determine $PV$.This revised structure and additional detail provide a unified introduction to the comprehensive methodology used in our study, ensuring that the complete process is clearly presented before delving into the individual parts.
The specific modifications are as follows:
The distinction from previous studies lies in that $I$, in the building susceptibility equations proposed by Li and Singh, represents landslide intensity. For Li, $I$ is influenced by the dynamic intensity factor ($Idyn$) and the geometric intensity factor ($Igem$), while for Singh, $I$ is determined by the landslide volume ($v$) and the expected velocity of the landslide ($s$). In this study, $I$ is innovatively redefined as landslide hazard, and a convolutional neural network is employed to quantify this hazard as a specific value within the interval from 0 to 1.
Data was collected from various sources, including satellite imagery, geological surveys, and historical landslide records. The collected data was then cleaned to remove any inconsistencies and standardized to ensure uniformity. To address the issue of class imbalance in the dataset, we applied the Synthetic Minority Over-sampling Technique (SMOTE).We employed a Convolutional Neural Network (CNN) to quantify the landslide hazard ($I$). The CNN model was trained on the preprocessed data to learn the complex relationships between the input features (such as topography, geological structure, vegetation cover, etc.) and landslide occurrences. The model architecture included multiple convolutional layers with 3x3 filters, ReLU activation functions, and max-pooling layers to extract and process spatial features. The output of the CNN model was a probability value between 0 and 1, representing the landslide hazard for each building.Building resistance ($R$) was quantified using six key factors: structural composition, number of floors, construction materials, building age, geographic location, and accessibility. Each factor was assigned a resistance coefficient based on empirical data and expert judgment. The overall building resistance was calculated using the geometric mean of these coefficients to ensure a balanced contribution from each factor.The vulnerability of each building ($PV$) was assessed by integrating the landslide hazard ($I$) and building resistance ($R$) using the proposed formula. This formula quantifies the vulnerability of over 230,000 buildings in Xinxing County, expressing each building's vulnerability as a specific value between 0 and 1. This detailed quantification allows for precise vulnerability mapping and targeted risk mitigation strategies.
4) Chapter 6 “Conclusions” should be completed by the numerical results, obtained in course of the current study.
Thank you for your detailed review and valuable suggestions regarding our manuscript. In response to your comment about Chapter 6 "Conclusions," we have revised this section to include specific numerical results obtained during the current study. The updated conclusions now provide detailed percentages and numbers of buildings categorized under different vulnerability classes and landslide risk zones, reflecting the accuracy and findings of our assessment.
Revised Conclusions:
This study presents a comprehensive assessment of building vulnerability to landslides on a large scale by integrating quantitative and machine learning methods, using Xinxing County in Guangdong Province as a case study. Initially, we collected extensive data on various factors influencing landslide occurrence, including topography, geological structure, vegetation cover, and land use. Using statistical methods, we identified key factors significantly impacting landslide risk. Subsequently, we employed a Convolutional Neural Network (CNN) model to evaluate landslide hazard risks and combined this with an assessment of building resistance based on structural and environmental parameters. This approach allowed us to calculate a detailed vulnerability value (PV) for each building.
The model can accurately and reliably predict the building vulnerability class in most cases, achieving an accuracy rate of 94.5% based on on-site validation. Specifically, the landslide hazard assessment categorized the county into four hazard zones: low, medium, high, and extremely high, covering areas of approximately 469.41 km², 444.44 km², 396.16 km², and 199.36 km², respectively. The vulnerability assessment identified that 18% of buildings are in low-risk zones, 7% in medium-risk zones, 17% in high-risk zones, and 24% in very high-risk zones. In terms of building vulnerability, the study found that 31.87% of buildings exhibit low vulnerability, 18.00% exhibit medium vulnerability, 34.23% exhibit high vulnerability, and 15.90% exhibit very high vulnerability.
By integrating machine learning with quantitative methods, this study provides a robust tool for disaster risk management, enhancing the ability to predict and respond to landslide hazards with greater accuracy. This approach supports the development of targeted mitigation strategies and contributes to the broader field of geohazard management with a scalable and adaptable assessment model.
Sincerely,
Fengting Shi
